# Implementation of a Pig Toilet in a Nursery Pen with a Straw-Littered Lying Area

**DOI:** 10.3390/ani12010113

**Published:** 2022-01-04

**Authors:** Michelle Tillmanns, Kees Scheepens, Marieke Stolte, Swetlana Herbrandt, Nicole Kemper, Michaela Fels

**Affiliations:** 1Institute for Animal Hygiene, Animal Welfare and Farm Animal Behaviour, University of Veterinary Medicine Hannover, Foundation, 30173 Hannover, Germany; Nicole.Kemper@tiho-hannover.de (N.K.); Michaela.Fels@tiho-hannover.de (M.F.); 2Educatijeboerderij Walnoot & Wilg, Oirschot, Foundation, 5688 GX Oirschot, The Netherlands; kscheepens@aol.com; 3Statistical Consulting and Analysis, Centre for Higher Education, TU Dortmund University, 44227 Dortmund, Germany; marieke.stolte@tu-dortmund.de (M.S.); swetlana.herbrandt@tu-dortmund.de (S.H.)

**Keywords:** pig, pig toilet, eliminative behaviour, animal welfare

## Abstract

**Simple Summary:**

Generally, pigs are known to be very clean and intelligent animals. The present study tested whether it is possible to train nursery pigs to defaecate and urinate in a pig toilet while keeping a straw-bedded lying area clean. The pig toilet was separated into a defaecation area and a urination area. An automatic rewarding system installed in the urination area and pig faeces placed in the corners of the defaecation area were used to help the pigs to identify the elimination area. By means of direct observation and video analysis, all eliminations within the experimental pen were analysed. Altogether, the pig toilet was very well accepted by the pigs, but a spatial separation of faeces and urine was not achieved. The lying area was kept clean, so that keeping pigs on straw in combination with a pig toilet would be quite conceivable in the future to enable a more animal-friendly life.

**Abstract:**

In this study, a pig toilet was installed on an organic pig farm, which enabled pigs to use a lying area littered with straw and keep it clean. The pig toilet was separated into a defaecation area and a urination area and nursery pigs were trained to use the urination area by means of a rewarding system. A total of 24 piglets were weaned at 6–7 weeks of age and housed in the experimental system for four-week periods. Per trial, a group of four pigs was formed, and videos were recorded on two days per week (08:00 to 18:00). Direct observation was carried out in the first and last week of each trial. In total, 1500 eliminations were video-analysed. An individual pig had an average of 7.1 ± 1.4 defaecations and 4.8 ± 0.8 urinations per day. In total, 96.4% of all urinations and 97.4% of all defaecations were performed in the pig toilet. However, most urinations took place in the defaecation area as well (90.4%). Even if the training to spatially separate defecation and urination behaviour was not successful, we showed that a pig toilet offers the possibility to create littered lying areas, possibly increasing animal welfare.

## 1. Introduction

Pigs are considered to be the cleanest farm animals kept by humans if the housing system provides sufficient space for them to live out their normal excretory behaviour [1]. It is their natural behaviour to leave the nest for defaecation and urination [2]. When pigs are kept in extensive housing systems, they leave their hutches to defaecate and urinate [3,4]. Different perspectives exist on whether this behaviour is innate or learned and which role the sow plays in the development of the piglets’ excretory behaviour. However, it was shown in studies on motherless rearing that the piglets also left the nest to eliminate [1,5]. With or without maternal rearing, it is a fact that piglets keep the nest clean independently at few days of age [6]. Consequently, pigs and even piglets classify their environment into functional areas, mainly activity, resting/lying and elimination areas [7,8]. In fact, there are existing studies that use this behaviour to improve the welfare of livestock, e.g., installing elevated platforms equipped with manipulable material to create an area for activity [9]. Once classified, the lying area is kept as clean and dry as possible by the pigs, and the elimination area is mostly located in corners of the pen or next to the walls [5,10].

However, if the pens are overcrowded, the normal excretory behaviour is impaired and the entire pen area may be soiled [10,11] resulting in more ammonia formation and emissions, especially when the floors are unslatted. In pens with slatted floors, low animal densities can also lead to higher ammonia levels since the pigs do not kick the manure through the slatted floor as quickly [12]. Ammonia is a volatile compound of nitrogen and is formed by the breakdown of urea from urine by the enzyme urease, which is naturally present in faeces [13]. The air pollutant itself and the formation of particulates compromise both human and animal health [14]. Confining numerous animals to a small space increases the ammonia levels, and stronger odours cause the indoor air quality to deteriorate. The conventional pig farming sector was accountable for 19% of ammonia emissions in Germany, and 95% of all ammonia emissions originated from agriculture (Umweltbundesamt, 2020). Accordingly, reducing ammonia emissions in livestock housing is not only of significant importance in the context of animal welfare and work safety but also in terms of environmental protection [15].

Currently, this problem is counteracted by installing air scrubbers [1]. These direct clean air into force-ventilated livestock buildings, filtering pollutants from the stale air and then returning it to the outside. This protects the environment but does not change the existing conditions for pigs in these housing systems. In order to improve the well-being and health of pigs and to make working conditions more attractive for farmworkers, pen soiling and ammonia formation have to be prevented. Therefore, the animals’ preference to separate areas for lying and excreting and to eliminate near walls and in corners could be utilised, and thus a functional area for excreting could be created [16].

Correctly used, such a pig toilet could represent an innovative solution for pig farming, especially when the system could be operated automatically, i.e., without increased labour input for the farmer. The pig toilet as a separate compartment within the pen would also make slatted flooring largely obsolete and thus offer the possibility of providing a straw-covered lying area for the animals. This would be a major step forward for pig welfare and could provide further environmental benefits, especially if urine and faeces were collected separately, since this could not only further reduce ammonia emissions but also provide valuable nutrients for fertilising fields. Accordingly, encouraging pigs to use such a designated area for urination and defaecation would improve air quality, opening up the possibility to provide a more comfortable floor to rest on, to supply manipulable materials for natural rooting behaviour and to reduce tail biting and perhaps provide more (outdoor) space and a wallow in the future [17]. Establishing a pig toilet in conventional pig farming could be an important and innovative step towards more sustainable pig production [13].

With regard to the use of operant conditioning in livestock to influence the location at which elimination behaviour is carried out, there are hardly any scientific studies available to date. Controlled elimination behaviour, as it is known in humans [18], can also be achieved in other mammals, such as dogs or cats [19], through associative learning paradigms in the form of operant and classical conditioning. For successful toilet training, the animals must learn to recognise the excretory stimulus, suppress it until they reach the toilet, eliminate and then leave the toilet again [20]. One way to achieve this is to reward the desired behaviour through positive reinforcement. This was already studied in cattle [20,21] but not yet in pigs.

Therefore, the aim of the present study was to convert an area spatially separated from other functional areas within a nursery pen into a pig toilet and to train nursery pigs to exclusively use this area for elimination. Our intention was also to achieve a separate defaecation and a separate urination area within the toilet to prevent ammonia formation. In the defaecation area of the toilet, feed was offered, while in the urination area, the drinker was located. It was hypothesized that the placement of the drinker in this area should induce urination there immediately after drinking (as proposed in the literature; [22]). These first “quasi random” correct urinations after drinking should be rewarded (by providing a food reward) so that the animals establish a positive link between urination and this area and will go there intentionally to urinate in future. In addition, the animals were provided with a lying area littered with straw, which, at the same time, should not lead to an increased workload for the farmer.

## 2. Materials and Methods

### 2.1. Experimental Design

The studies were conducted in the years 2018 to 2020 on an organic pig farm in Oirschot, North Brabant, The Netherlands. A housing licence existed for 50 sows, but during the trial period only 16 to 20 sows were kept on the 10-hectare farm. Most sows were of the Berkshire breed, the oldest pig breed in Great Britain, but there were also some hybrids because of two Angler saddle sows kept on the farm as well. No surgical interventions such as castrations, tail docking or teeth grinding were performed on this farm. Neither iron injections for piglets, which are common in commercial farming, nor common vaccinations for suckling piglets such as those against *Mycoplasma hyopneumoniae*, ileitis, PRRS and/or circoviruses (PCV-2) were performed. The pigs on the farm were usually kept outdoors all year round on adjacent 750 m^2^ parcels, which were equipped with metal huts littered with straw. However, for the present study, pigs were kept in an experimental indoor pen after weaning and the experiment started immediately. The pigs were weaned at the age of six or seven weeks. All pigs on the farm were fed once a day with an organic complete feed for pregnant sows (Reudink B.V., Lochem, The Netherlands) and other organic feed, for example organic bread.

The full feed composition is listed in Table 1. The meat of these pigs was offered via selected partners, to support the concept of this welfare-friendly system.

For the present study, on this farm, an experimental pen was constructed which was equipped with a pig toilet. During the study period, a total of 24 nursery pigs (14 females and 10 males) were kept in this pen. Therefore, at 6 to 7 weeks of age, the experimental animals were weaned and brought over from the outdoor area to the indoor pen, which was located in a small barn. In each experimental cycle, four littermates were housed together in the experimental pen for a total period of four weeks. The pen consisted of a flat floor area (12.4 m^2^) generously littered with straw and a pig toilet (2.1 m^2^), which was separated from the littered area on three sides by 1 m high opaque walls (Figure 1 and Figure 2). Inside, the toilet was divided by an opaque partition (height: 1 m) into a defaecation area and a urination area. The pig toilet had a slatted iron floor with 1 cm wide gaps and equally large steps and a manure channel underneath. The urination area of the toilet was equipped with an automatic rewarding system to train the animals to urinate there using operant conditioning. Under the urination area, there was a metal tank with a connected reservoir with two electrodes in it. Once the reservoir was filled with 20 mL of liquid (i.e., urine), the electrodes generated an electrical potential which firstly opened a flap on the reservoir, making the system refractory for 30 s, and secondly delivered a reward via the automatic rewarding system. The automatic rewarding system consisted of a metal box that could be covered with a lid. The box included a food snail, which rotated as soon as an electrical potential was generated. Inside the snail there were biological candies and biological sugar, which prevented the candies from gluing together. Through a tubular system, the reward was dispensed into an aluminium bowl, which was located at the front of the urination area next to an overflow drinker. This was the only drinker in the pen; thus, the animals had to go to the toilet to drink. The candies that were used for the experiments were originally produced for human consumption; thus, it was not pig feed. For possible later practical use, the candies would have to be approved as feed for pigs.

Immediately beside the entrance to the defaecation area of the toilet, there was a feeder with two feeding places for ad libitum feeding of the pigs with a biological complete feed (Figure 1). This was the same feed which the piglets already received during rearing when they were outside (organic complete feed for pregnant sows, Reudink B.V., Lochem, The Netherlands, Table 1). It has to be emphasized that the feed composition used on the farm was designed to meet the nutritional requirements of sows and is not the optimal feed for piglets. Additionally, candies are not approved as feed for pigs. Thus, this feed composition was used for the experiments, but it is not necessarily suitable for practical use. Before the weaner pigs entered the experimental pen, the corners of the defaecation area were equipped with fresh pig faeces previously collected in the outdoor area where the piglets grew up so that they could identify the elimination area. The pig toilet was conceived by the pigs’ owner and the implementation of the rewarding system was realised with the support of the company Kamplan B.V., Boxtel, The Netherlands.

On the first day of each experimental cycle, the pigs were weighed, sexed and ear tagged. Furthermore, the animals were individually marked (a small area on the back was sheared and marked with an animal marking pen) for better identification. The latter procedure was repeated several times during the experimental cycle when the colour had worn off or the bristles of the animals had grown back. On the last day of the experimental cycle (i.e., after four weeks), the pigs were weighed again, taken from the pen and integrated into a single sex group in the outdoor area.

### 2.2. Direct Observation of Excretory Behaviour

To analyse the excretory behaviour of the pigs and their use of the toilet, direct behavioural observation was performed during the first week and the last three days of an experimental cycle. It was initially conducted to record events that might be missed by video analysis (please see Section 2.3) and thus obtain more accurate data. Furthermore, it was possible to determine whether information gained from the video and direct observation were comparable.

For direct observation, the observer stood approximately 2.5 m away from the experimental pen. A total of five experimental cycles with four animals each, i.e., 20 pigs, were included in the evaluation via direct observation. For the behavioural analysis, a monitoring protocol was prepared and the data were initially written down by hand. The observation period covered 10 h per day, from 08:00–18:00. During this time frame, the animals were observed continuously. Each drop of faeces and urine was documented and it was determined which animal eliminated at which time in which area of the pen. Thus, it was noted whether an individual urinated or defaecated in the lying area, in the defaecation or urination area of the toilet or in the direct vicinity of the toilet.

### 2.3. Video Analysis

Three cameras (EQ900F EZ.HD 720 pixel cameras, EverFocus^®^, New Taipei City, Taiwan with YV2.8x2.8SR4A-SA2L 2.8 mm–8 mm zoom lenses, FUJINON Lens, Fujifilm, Tokyo, Japan) were installed in the experimental pen. The cameras were attached 2.20 m above the lying area, the defaecation area and urination area of the pig toilet and connected to a digital video-recorder (Monacor AXR-108 8-channel HYBRID, MONACOR INTERNATIONAL GmbH & Co.KG, Bremen, Germany). Hard drives (Western Digital WD Purple WD20PURZ 3.5-inch, internal, SATA III, San Jose, CA, USA) with 2TB or 3TB storage space were used to record the videos. Using a portable DVB-T player (Lenco^®^ TFT-1026 25.5 cm, 10.1 inch, DVB-T, HDMI, USB, Nuth, The Netherlands) the recorded videos were transferred from the internal hard drives to external 3TB or 8TB hard drives (Intenso Memory Center GmbH-Hard Drive-External (Stationary)-3.5 (8.9 cm)-USB 3.0, Vechta, Germany), which were connected to a personal computer for analysing the videos. The video footage was fully reviewed, and two consecutive days per experimental week when there was little distraction for the pigs in the barn were selected and analysed between 08:00–18:00 using VLC media player.Ink (3.0.11). The behavioural analysis was carried out in a similar manner to the procedure for direct observation. Due to technical reasons, video evaluation was performed in four experimental cycles, i.e., for a total of 16 pigs. A summary of the experimental cycles is shown in Table 2.

### 2.4. Statistical Analysis

Statistical analyses were carried out using the statistic software R [23]. At first, data from direct and video observation were analysed descriptively and frequencies and percentages of defaecations and urinations in different pen areas were calculated per individual pig. A logistic regression model was used for urination (1 corresponded to urination in urination area, 0 corresponded to urination in lying/defaecation area) and defaecation (1 corresponded to defaecation in defaecation area, 0 corresponded to defaecation in lying/urination area) separately, with observation day, sex and body weight as fixed effects and the experimental cycle, the individual pig and the observation method as random effects [24]. The aim of this analysis was to find out if correct urination in the urination area and correct defaecation in the defaecation area of the toilet were affected by different factors such as observation day, sex or individual body weight of pigs. The factors body weight and sex were included in the statistical model since there was no information from literature how body weight and sex can affect toilet training success for urination.

## 3. Results

### 3.1. Frequency of Urination and Defaecation per Day and Animal

A total of n = 888 defaecations and n = 612 urinations were recorded in the video analysis, while in the direct observation there were n = 576 defaecations and n = 335 urinations. Table 3 illustrates the daily frequency of the urine and faecal excretion of individual animals in different experimental cycles obtained by video analysis.

### 3.2. Use of the Pig Toilet in Direct and Video Observation

The results of the direct observation revealed that 99.2% of all observed defaecations were located in the defaecation area of the pig toilet. Only 0.8% of all defaecations took place in the lying area and in the urination area of the toilet. The defaecation area was also used for 78.1% of the pigs’ urinations. In the actual urination area, urine was deposited with a frequency of 21.2% and 0.7% of the urinations occurred in the lying area. For technical reasons, only data for 16 animals from four cycles were available for the video analysis. The video analysis showed that the percentages of defaecation in the lying and urination area was slightly higher than during the direct observation (2.6% and 0.8%, respectively). In addition, the percentage of urination was higher in the lying area (3.6%) and lower in the urination area (6%) compared to the results of the direct observation. However, also in the video observation, almost all urinations (90.4%) occurred in the defaecation area of the pig toilet (Figure 3).

### 3.3. Excretory Behaviour of Individual Animals Related to Video Analysis

The observation of the excretory behaviour of individual animals during the four cycles of video analysis revealed that most animals performed nearly 100% of defaecations in the defaecation area of the pig toilet (Figure 4A). However, there were also some differences between individual animals and experimental cycles. For pigs with the numbers 5–8 from the second cycle, it was shown that the percentage of defaecations in the lying area was higher than for the pigs of the other cycles; the highest percentage achieved was 12.2% for pig number 7 (Figure 4). Additionally, this group had the highest percentages of correctly excreted urine in the urination area of the pig toilet, with 40% (animal 6) and 42.2% (animal 7). At the same time, however, it was noticeable that the percentage of urinations in the lying area was the highest for these animals in relation to all the other animals in different experimental cycles and that this was only exceeded by the two animals from the same cycle.

### 3.4. Utilisation of the Pig Toilet over Time

The defaecation area of the pig toilet was used for both digestive finals and micturition, so that excretion mainly took place in this area. The highest percentage of correct defaecation in the defaecation area was observed during the first 10 days and the last week of an experimental cycle (Figure 5A). The percentage of correct urination in the urination area fluctuated over the entire observation period with the highest success during the first five days and at days 22 and 23 of observation (Figure 5B).

### 3.5. Results of the Statistical Model

The results of the statistical model revealed that correct defaecation and urination in the respective areas of the pig toilet were not affected by the sex and body weight of the pigs (*p* > 0.05). For defaecation, a significant effect of the observation day was found (*p* < 0.001). The probability of faeces being correctly deposited in the defaecation area decreased by a factor of 0.907 when the observation day was increased by one day (Table 4). For urination in the urination area, no significant effect of the observation day was found (*p* > 0.05, Table 5).

## 4. Discussion

In the present study, a pig toilet was established in a pen for weaner pigs and their excretory behaviour was analysed in detail. Per ten-hour-observation day, an individual piglet had 7.1 ± 1.4 defaecations and 4.8 ± 0.8 urinations on average during video observation. Thus, we observed 11.9 eliminations per piglet and day in total. In previous studies, the excretory behaviour of suckling piglets or fattening pigs was investigated, whereas, to the best of our knowledge, there is hardly any information on weaner pigs from previous studies. Buchenauer et al. [10] observed 16.4 eliminations on average per piglet within the first 24 h of life. For six-week-old weaner pigs, four to seven defaecations per day (24 h) were reported [25], while no information on the number of urinations is available. According to Guo et al. [26], fattening pigs had 17.9 ± 0.32 eliminations per animal and day, on average. They observed that 67% of all excretions occurred during the daytime between 06:00 and 18:00, which was very similar to the observation period of the present study (08:00 and 18:00). In the study by Guo [26], the amount of 67% of all excretions corresponded to a total of 11.9 eliminations per animal occurring between 06:00 and 18:00. This is completely in line with the results of the present study, indicating that the number of excretions per day is similar for fattening and nursery pigs. Pigs mostly eliminate during the daytime, and the reduced number of excretions at night is probably related to the decreased activity of the animals. Various authors confirm that the activity of pigs decreases with the beginning of twilight [27,28,29].

The observation that pigs divide their environment into functional areas and that these are consistently respected was confirmed in the current study. Consequently, the straw-covered area of the pen was mainly used for activity, resting and rooting behaviour, while the pig toilet was used for the purpose it was made for, with eliminations mainly taking place in the defaecation area of the toilet. Feeding took place in this area as well, because the automatic feeder was installed there. The idea of combining feeding and the defaecation area was based on the deep-litter-pen model presented by Mayer et al. [30], where the littered lying area was located lower than the elimination area and the latter was equipped with a feeder. As in the studies by Whittemore [31] and Wang et al. [32], we used the pigs’ faeces to mark the defaecation area of the pig toilet. Andersen et al. [33] observed that sows that were fixed in crates stretched their heads as far as possible away from the feeder and the lying area while excreting, which could be evidence of the animals not wanting to excrete in the feeding area, though they were not able to do otherwise. However, during the experimental period in the present study, it was not observed that the pigs tried to move away from the feeding area for excreting. On the contrary, they immediately accepted the defaecation/feeding area and almost all eliminations were carried out there. A control group could provide more information about the excretory behaviour of pigs when no toilet is provided in the same pen. However, due to the conditions on the farm (number of animals, structural conditions), we were unfortunately unable to integrate a control group into the study. In the area of the toilet that was intended to be used for urination, almost no excretory behaviour was observed. The animals entered this area for drinking but hardly ever for urination. According to Hacker et al. [22], the overflow drinker and the humid environment should have encouraged urination in this area, which was not the case here. One reason for the poor use of the urination area could be the structural conditions, as with a width of 44 cm the urination area seemed to be rather narrow; the animals could only turn around with difficulty and had to leave the area backwards with increasing size. Since a visit to the urination area and an initial accidental urination there, which is rewardable, is the prerequisite for successful operant conditioning, toilet training may have been made more difficult by the structural conditions of the pig toilet—or was even not possible. Furthermore, the function of the rewarding system needs to be critically questioned. It was observed that the animals quickly became used to the sound of the opening flap and the sound of the rotating snail of the rewarding system and rushed into the urination area to catch the reward, so that an animal that had just urinated might not receive a reward because it had been stolen by a penmate. Furthermore, the rewarding system was sometimes triggered several times so that the system was inactive thereafter for a short time but then immediately activated again, dispensing a new reward. This could indicate that the system reacted too sensitively and that a reward was already triggered with a smaller amount of liquid than 20 mL. It is possible that, due to this situation, the pigs were not able to associate the urination with a subsequent reward. Nevertheless, there were a few pigs that urinated more frequently in the urination area than others. This may indicate individual differences in the learning behaviour of the animals. Statistically, no effect of the sex or body weight of the pigs was found in relation to correct defaecation and urination. It became obvious that individual differences in the use of the pig toilet were particularly evident in the second cycle of the experiment, so it could be assumed that different circumstances existed between the cycles or that the animals influenced each other’s behaviour. Specifically, it was noticeable that only in the second experimental cycle urine was deposited in the littered area by two animals (animals 6 and 7), and only animals from the first and second cycle deposited urine in the urination area of the pig toilet (animals 1 and 3 from the first and animals 5, 6, 7 from the second cycle). There was no change in the experimental design or the function of the rewarding system in the third and fourth cycles. Thus, it remains unclear why the pigs did not use the urination area in these cycles, especially since they accepted the toilet as such and all eliminations took place in the defaecation area of the toilet.

The results of the logistic regression models for the variables defaecation (correct/wrong) and urination (correct/wrong) revealed that there was a significant effect of the observation day during the housing period on the correct defaecation in the defaecation area of the toilet, which was not the case for urination in the urination area. However, the effect of the observation day concerning defaecation must be critically questioned. Accordingly, the probability of faeces being deposited in the defaecation area of the pig toilet would be reduced with increasing observation days. Nonetheless, we descriptively showed that there were only some differences between the 10th and 25th observation day, with slightly reduced use of the defaecation area compared to the other observation days. Between the 25th and the 30th day of observation as well as in the first 10 days, the defaecation area was intensively used by the pigs. This could also be a reason why, in the direct observation, higher percentages for correct defaecation and urination were detected compared to video analysis, since the direct observation was carried out in the first week and at the end of the experimental cycle. Overall, it became evident that the pig toilet—more precisely the defaecation area—was used continuously by the pigs during the entire four-week housing period.

The use of a pig toilet could be reduced at high temperatures in summer when pigs avoid lying in straw and use the straw-bedded lying area for eliminative behaviour. Considering that, one could assume that the second experimental cycle of the present study took place in summer, since the pigs eliminated more often in the lying area than in the other experimental cycles. However, the reason for this remains unclear because the second cycle was carried out in winter and there was no effect of high temperatures during that time. Nevertheless, housing facilities that provide straw as bedding material should have access to adequate temperature control. The farm on which the present study was performed did not have any facilities to reduce the barn temperature, but in this regard installing of air conditioning is being considered. Nannoni et al. [17] also mentioned in their scientific review their vision of animal-friendly, straw-using housing systems with pig toilets and additional wallowing facilities. However, the use of straw may also have hygienic disadvantages, for example, straw varies in quality and can be a medium for bacteria and fungi. The incompatibility of manure removal systems such as slurry channels and the use of slatted floorings in combination with straw are further disadvantages regarding the use of straw in pig farming [34]. The latter problem could be avoided if the installation of a pig toilet would make it possible to dispense with slatted floors and offer a straw-bedded lying area that is kept clean by the pigs. The present study shows that this is fundamentally possible. Even if the division of the toilet into faecal and urine areas did not lead to the desired success, the pigs accepted the toilet as such and the majority of the eliminations took place there. Since the straw-bedded lying area was kept clean, no manure removal was necessary during an experimental cycle.

## 5. Conclusions

The present study revealed that, in principle, it is possible to establish a pig toilet for nursery pigs kept in small groups. A well-functioning pig toilet enables the animals to live out their natural behaviour and to classify their environment into functional areas. A flat-surfaced lying area littered with straw can be provided, with the aim of increasing animal welfare, if care is taken to ensure that the selected straw is hygienically safe. The pig toilet in the tested system did not facilitate a separation of the urination and defaecation behaviour. It is conceivable that an architectural revision of the prototype pig toilet in this study could be more successful with regard to the separation of faeces and urine, having the potential to reduce ammonia emissions. Overall, more research is needed in this field, especially in relation to conventional housing systems with larger groups of pigs.

## 6. Patents

The second author (KS) owns a patent with the patent number PCT/NL2014/050856 for the pig toilet equipped with the operant conditioning system for rewarding urination behaviour used in the study.

## Figures and Tables

**Figure 1 animals-12-00113-f001:**
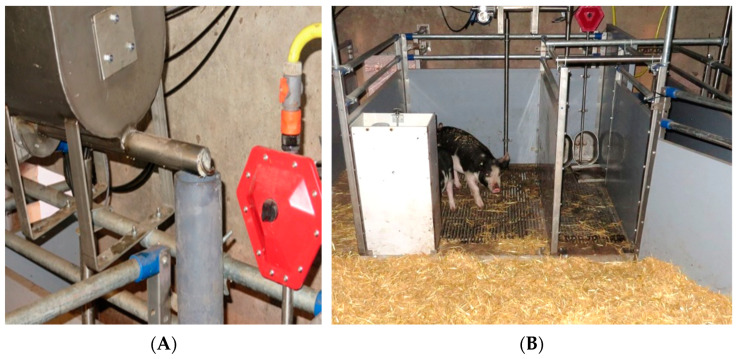
(**A**) Automatic rewarding system consisting of a tank for the organic lemon candy and organic sugar, feed snail and pipeline system above the urination area of the pig toilet; (**B**) pig toilet with defaecation area (**left** side) and urination area with overflow drinker and rewarding system (**right** side).

**Figure 2 animals-12-00113-f002:**
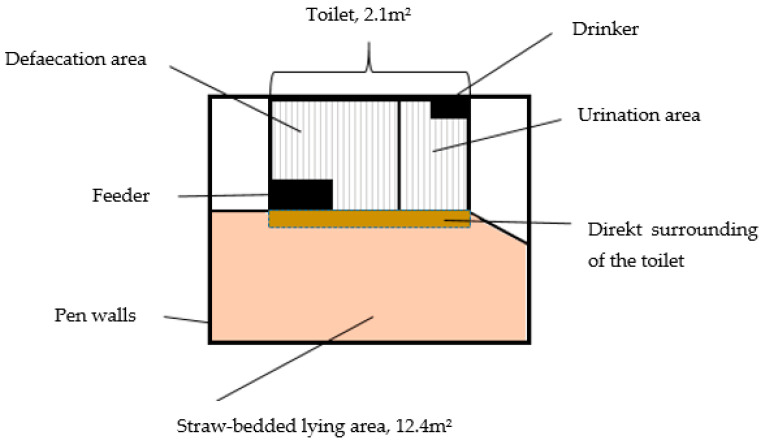
Experimental set-up.

**Figure 3 animals-12-00113-f003:**
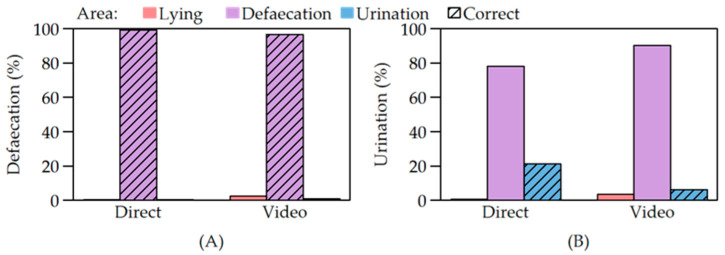
(**A**) Percentages of defaecations; (**B**) percentages of urinations in the lying area, defaecation area and urination area of the pig toilet depending on the observation method (direct observation or video analysis).

**Figure 4 animals-12-00113-f004:**
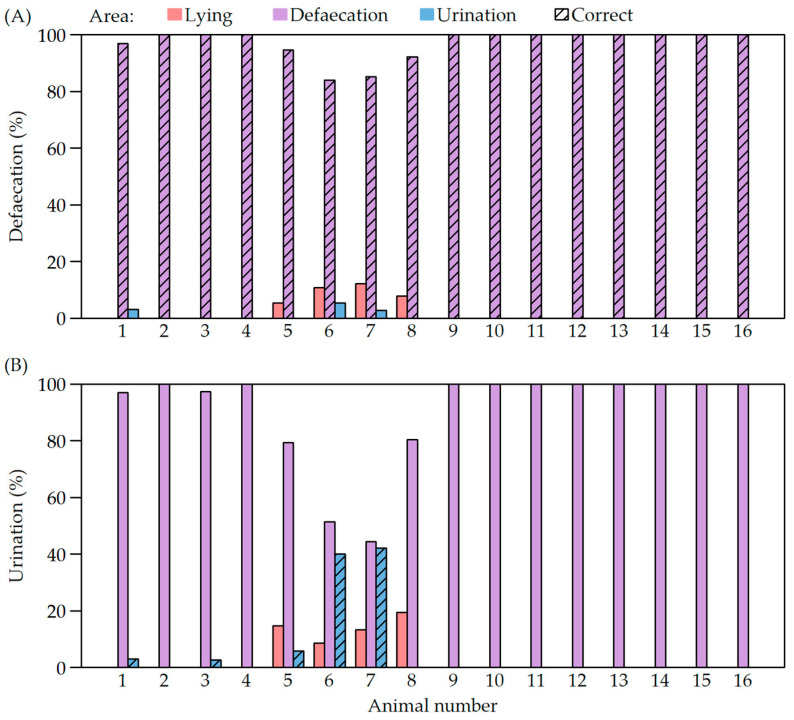
(**A**) Percentages of defaecations and urinations of individual animals during video observation; (**B**) percentages of urinations of individual animals in the different functional areas (lying area, defaecation area and urination area of the toilet) during video observation.

**Figure 5 animals-12-00113-f005:**
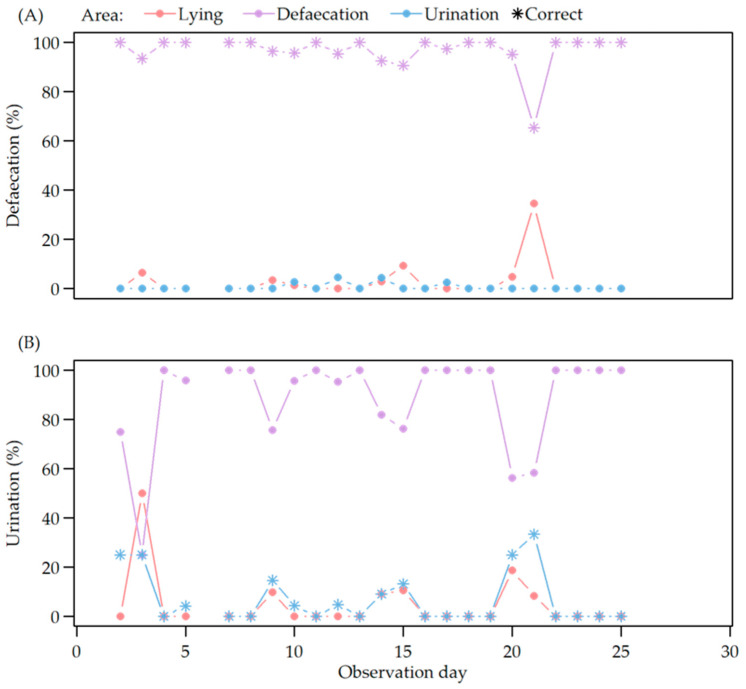
(**A**) Defaecation in relation to observation day during video observation; (**B**) urination in relation to observation day during video observation in the different areas of the pen (lying area, defaecation area and urination area of the toilet).

**Table 1 animals-12-00113-t001:** Composition of the Reudink B.V. (Lochem, The Netherlands) organic complete feed for pregnant sows.

Composition: Dry matter content: 97% with 97% of combined agricultural origin, of which 84% from organic raw materials, 15% from in-conversion raw materials and 0% from regular raw materials. Feed from mineral and other sources: 3%. Wheat flour, wheat gluten, peas, beet pulp, maize meal, organic sunflower seed hulls, organic alfalfa, organic oats, organic wheat starch, organic molasses, calcium carbonate, monocalcium phosphate, sodium chloride, potato protein.
Analytical constituents: Crude protein 13.7%, crude fat 4.5%, crude fibre 8.6%, crude ash 6.1%, lysine 0.64%, methionine 0.20%, calcium 0.73%, phosphorus 0.73%, sodium 0.22%
Supplements (per BW): Vitamins: Vitamin A (3a672a) 12,000 IU, Vitamin D3 (3a671) 2000 IU, Vitamin E (3a700) 80 IU. Trace elements: Cu(3b405) copper(Il)sulphate, pentahydrate 15 mg, Fe(3b103) iron(l)sulphate, monohydrate 150 mg, 1(36202) calcium iodate anhydrous 2.00 mg, Mn(3b502) manganese(Il)oxide 40 mg, Se(3b801) sodium selenite 0,30 mg, Zn(3b605) zinc sulphate monohydrate 65 mg.

**Table 2 animals-12-00113-t002:** Summary of performed and analysed experimental cycles.

Cycle	Animal-Nr.	Sex	Analysis
1 (October 2019)	1	female	direct
	2	male	direct
	3	female	direct
	4	female	direct
2 (November 2019)	5	male	direct + video
	6	female	direct + video
	7	female	direct + video
	8	male	direct + video
3 (December 2019)	9	female	direct + video
	10	female	direct + video
	11	female	direct + video
	12	male	direct + video
4 (June 2020)	13	male	video
	14	female	video
	15	female	video
	16	male	video
5 (July 2020)	17	male	direct + video
	18	male	direct + video
	19	male	direct + video
	20	female	direct + video
6 (September 2020)	21	male	direct
	22	female	direct
	23	female	direct
	24	male	direct

**Table 3 animals-12-00113-t003:** Number of defaecations/urinations per day for all individuals in different experimental cycles (means, standard deviations, minimum and maximum) obtained by video analysis.

Cycle	Animal-Nr.	Defaecations per Day (Mean ± SD)	Min. Defaecation	Max. Defaecation	Urinations per Day (Mean ± SD)	Min. Urination	Max. Urination
1	1	8.3 ± 3.7	3	14	4.3 ± 1.7	2	7
	2	6.3 ± 3.9	0	10	4.6 ± 1.7	3	7
	3	6.8 ± 3.9	1	12	4.8 ± 1.5	2	7
	4	6.8 ± 3.4	2	11	5.0 ± 0.5	4	6
2	5	9.4 ± 3.5	4	13	4.3 ± 1.9	1	7
	6	7.0 ± 2.6	4	12	4.4 ± 1.6	2	7
	7	9.3 ± 3.7	6	16	5.6 ± 3.9	1	13
	8	8.5 ± 2.6	6	12	6.8 ± 2.1	4	10
3	9	6.6 ± 2.1	4	10	5.6 ± 2.3	3	10
	10	4.6 ± 1.1	3	6	5.3 ± 1.8	3	7
	11	5.1 ± 1.6	3	7	5.0 ± 1.4	3	7
	12	5.6 ± 2.3	2	9	6.0 ± 2.8	2	11
4	13	6.5 ± 2.5	1	9	3.9 ± 1.4	1	5
	14	6.5 ± 1.9	3	9	3.8 ± 0.5	3	4
	15	7.5 ± 2.9	3	12	4.8 ± 2.1	1	7
	16	8.5 ± 3.0	5	13	4.3 ± 1.7	2	7

**Table 4 animals-12-00113-t004:** Results of the logistic regression model for defaecation.

	Estimate	Odds Ratio	Standard Error	z-Value	*p*-Value
Intercept	14.63		6.70	2.20	0.03
Sex (male)	−0.86	0.42	1.00	−0.87	0.39
Observation day	−0.10	0.91	0.03	−3.78	<0.001
Body weight (kg)	−0.23	0.80	0.23	−0.98	0.33

**Table 5 animals-12-00113-t005:** Results of the logistic regression model for urination.

	Estimate	Odds Ratio	Standard Error	z-Value	*p*-Value
Intercept	−5.40		3.95	−1.37	0.17
Sex (male)	−1.76	0.17	1.25	−1.41	0.16
Observation day	−0.01	1.00	0.02	−0.28	0.78
Body weight (kg)	0.07	1.07	0.18	0.36	0.72

## Data Availability

The data presented in this study are available on reasonable request from the corresponding author. The data are not publicly available due to privacy.

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
