# Peer review of "Implementation of a Pig Toilet in a Nursery Pen with a Straw-Littered Lying Area"

_animals, 2022, doi:10.3390/ani12010113_

Round 1

Reviewer 1 Report

Dear authors,

thank you very much for carrying out this study, which is basically very interesting! In my opinion, the results can be published as they are for a first study after improvements have been made, but further studies would be desirable in the future, especially in conventional husbandry systems and with more animals. The research group may have further data available here, or may be able to collect these in the near future as a supplement. Enclosed is my critique in detail:

Line 50:
This statement may be true for unslatted floors, but for a slatted floor the opposite is more likely to lead to higher emissions: if there are few animals in a large pen, the cadence is reduced and the manure does not get under the slatted floor as quickly. Therefore, this statement is not universal - please adjust.

Line 117-122: 
It is nice that the farm has such marketing. However, it sounds in these lines as if "advertising" should be done for this label. I do not think this is appropriate for a scientific article. Please delete or formulate very neutrally, as it has no relevance to the study itself.

line 141:
What "candies" were we talking about here? Are these on the positive list for feed? If not, has their use in the experimental animals been approved? How was it ensured that all animals perceived the intake of these sweets as a reward (relevant for the functioning of the reward system)?

Line 150:
Exactly what food and composition was used? Details are missing. This may have an effect on fecal consistency and defecation frequency.

Chapter 2.2: 
why exactly was a direct observation also carried out? As a reader, I wonder why everything was not recorded and evaluated purely via camera? In my opinion, the animals do not feel unobserved and do not show their "true" behavior. In practice, unfortunately, no one can stand next to them in the barn all the time so that the animals defecate and urinate "correctly" for us humans. The later conclusions may therefore be too positive.

Line 193: 
Tip: there is also special software that can simplify the evaluation of such studies. E.g. Mangold interact

Figure 3: 
The color change irritates the reader. I know what you want to say with the color change in defaecation and urination between the two diagrams, but I would prefer a uniform color scheme, possibly with hatching to highlight the "right" or wrong area.

Line 376: 
Thank you for this important advice! We must not ignore this in any discussion about animal welfare.

Line 406: 
Maybe this is the reason why some sentences sound like advertising... promotion by foundations affiliated with business enterprises unfortunately always has an aftertaste. Please check the entire text and write more neutrally at the appropriate places.

Author Response

Hello Reviewer 1,

Kind regards,

Michelle Tillmanns

Reviewer 2 Report

Simple summary:

The authors aimed to train pigs (or piglets?) to defaecate and urinate within two separate areas within the animals’ temporary home pen which combined functioned as “pig toilet”. The toilet training aimed to facilitate waste management and to keep the rest of the home pen free from excrements. To facilitate the toilet training, pigs received a food reward upon urinating in the correct urination location but defaecating in the correct area was not rewarded. The pigs performed almost all excretory behaviours within the toilet area although separation of urination and defaecation behaviours within the specified area was not achieved. The authors concluded that the implementation of the pig toilet was successful.

Overall, the manuscript is well structured and details to the experimental design well described, although at parts too extensive and not strictly relevant to the study. I have pointed out sections in the detailed comments below where I think is the case.

Main critic point:

My main critic is that the aim of animal training (i.e. using positive reinforcement to encourage excretion behaviour was not actually addressed. Data collection consisted of recording excretory behaviour but how the pigs used the reward system was not described nor did the authors link their observation of excretory behaviour with reward expectancy/use of the reward delivery system. Therefore, I argue that the authors cannot draw any conclusions whether the pigs were using the toilet because they were in fact “trained”. Maybe the pigs used the space correctly instinctively (i.e. without being place-conditioned) based on their natural tendency to use separate areas for excretory behaviours and other activities (as the authors describe very well in the introduction) when they have the space to do so. This notion is supported by the fact that almost all excretory behaviours were observed in the designated area from the start of the experiment. If the animals were not instinctively using the designated area for excretory behaviour, urination/defaecation within the entire pen would have been expected at the beginning of the experiment. The lack of a control condition (i.e. without the toilet apparatus in place) therefore limits to conclude whether the toilet, especially the reward mechanism, worked. Providing animals with adequate space might be enough to achieve the same results. Moreover, pigs were rewarded for urinating but not defecating, so how were they meant to learn to separate both areas with different excretory behaviours?

Introduction:

Overall, the introduction is very detailed with rewards to pigs’ natural behaviour and highlights the need for designing “pig toilets” from an animal welfare perspective and environmental reasons. However, background information to animal training and explanations to place conditioning using positive reinforcement is completely missing. Toilet training has been studied in other farm animals (e.g. cattle: Dirksen N, Langbein J, Schrader L, Puppe B, Elliffe D, Siebert K, Röttgen V, Matthews L. Animals (Basel). 2020 Oct 15;10(10):1889; Dirksen, N.; Langbein, J.; Schrader, L.; Puppe, B.; Elliffe, D.; Siebert, K.; Röttgen, V.; Matthews, L. How Can Cattle Be Toilet Trained? Incorporating Reflexive Behaviours into a Behavioural Chain. Animals 2020, 10, 1889.) and should be mentioned in the context of the current study.

Methods:

Although the authors state that the animals did not undergo any invasive handling procedures, I would have expected to find an official license number from their AWERB either in the methods or the institutional review board statement. Moving animals into a novel area and repeated handling during shearing/marking the animal for the purpose of the study (L160-163) is likely to be associated with stress. Moreover, safeguarding animal welfare in conditions likely to increase aggression associated with food competition (only one animal could receive food treats from the reward delivery apparatus in the toilet area; also the authors mention food competition issues in their discussion L320) should have been reviewed.

Given that the experiment started when the pigs were weaned, L105-111 describing housing of the parental generation seems irrelevant. Also, please make sure to be consistent in referring to the animals either as piglets or pigs throughout the manuscript. The latter seems more adequate given their age.

L119 – 121 explanation of meat quality not relevant to study.

L124 The total number of pigs is stated as 24 but number of animals in Figure 4A and B is 16, were the results for 2 groups (i.e. 8 animals not reported?).

L178 It would be good to know why the authors switched from direct to video observation. I assume this was done to safe time or more convenient but this should be made clear in the text.

Results:

It is stated in the methods that operant conditioning is used but without further details as to how this was actually achieved and also I would have expected to see any results on how well/quickly the animals learnt to use the reward system. Given that the pigs were not reward to defecate in a specific area, how was discrimination training of urination and defecation behaviours achieved?

Table 2 I’m not sure if this table adds any valuable information unless there is a reason to know how many times certain animals urinated/defecated (i.e. did some animals urinate more often, hence received more rewards and were therefore better in discriminating between urination and defection area?)

Figure 4. Why is the animal number 16 not 24?

L261 I don’t understand why the authors used a statistical model (which is a statistical procedure to explain variance) given that there is hardly any variance in their descriptive results. Moreover, it should be justified why sex and body were of interest here. Is there any evidence that sex and body weight influence excretory behaviour or learning?  

Discussion:

L 296-300 move to methods as this explains the design of the toilet and should be mentioned earlier on in the manuscript.  

L304 it is possible that the location of excretion behaviour was completely incidental given that the animals spent a significant amount of time in the area to feed?

L320 should discuss what the lack of reward delivery could have on conditioning the animals to use the toilet.

L326 this is very interesting and an important point to discuss. Judging from Figure 4 ,there were great difference between animals 5-8 (i.e. on particular group) in comparison to the others which could have been discussed here too (if this is what you are referring to in L 332-338, might be more clearer to refer to the results by animal number instead of cycle given that your figures refer to animal numbers)

L355 and below concerning straw discussion. Generally interesting but not sure how this relates to the current study given that the authors did not test the effect of straw bedding directly (i.e. groups with and without straw were observed).

L362-366 like above, I think an interesting point, but if I’m correct, only excretory behaviours were recorded in the toilet and the results do not show if the pigs really spent time lying in the toilet area.

Author Response

Hello Reviewer 2,

Kind regards,

Michelle Tillmanns

Reviewer 3 Report

General: This is a well readable, clearly written paper on a relevant subject.

  • The hypothesis tested whether the environmental impact can be decreased by separating defaecation behaviour and urinating behaviour is actual and relevant.
  • The data collected on frequency (and individual variation) of excretion behaviour is interesting, as these are not abundantly available for recently weaned pigs

I have a major comment: In fact, the aim of the experiment failed: the pigs did not separate both excreting behaviours. Authors should be more explicit on this in the abstract (!) and conclusions. The conclusion that pigs did separate function area’s (‘accepted the toilet’) is an open door in animal science and animal practice. To be fair regarding their aim, authors should amend their conclusion towards the (innovative!) aim of the study.

Now: “We conclude that installing a pig toilet offers the possibility to create littered lying areas in pig farming, possibly increasing animal welfare” should be “We conclude that installing a pig toilet in the tested system does not facilitate a separation of the urination and defaecation behaviour.” The text of littered area having possibly advantages for animal welfare has nothing new in it regarding pigs, which, also according to the authors, are sober animals.

Nevertheless, the data on the ethology of excretion behaviour are interesting and to some extend relevant. However, the authors have elaborated considerably on a relatively small data set, but omitted the opportunity to derive some relevant behavioural features such as 1) comparing defecating and urination behaviour (is there some degree of synchronicity in this?); 2) pattern over the day of both behaviours’ 3) sec effects et cetera.

The authors are presenting the farm as if they were writing a commercial for an organic farm. This should be more distant, especially now that the owner of the farm is co-author. Also, his patent doesn’t need to be highlighted to his person in the text (L153-155), as it is acknowledged in section 6 (Patents).

Line 116-117 “ which helps to minimise the 116 carbon footprint and the use of farmland for the cultivation of pig feed”  and Line 117-121: “Under the brand 117 name "Duke of Berkshire®", the meat of these pigs was offered via selected partners, such 118 as "De Groene Weg" or "Berkshire Butcher". The concept is to provide meat of excellent 119 quality and flavour, from animals that have been raised in a welfare-friendly system, and 120 is very well received by customers, even though the production costs and therefore the 121 final price for the consumer are higher”. These sentences are entirely irrelevant and should be removed.

The discussion on environmental temperatures on the outcomes (line 362 – 372). For this reason, I strongly advise to add month, year en ambient temperature at pig level in Table 2.

Statistics seem well organised. Why is no clear comparison between direct observation and camera observation in cycle 2, 3 and 5? Line 218 – L229 provides quite some motivation for this. Such a comparison would validate the methods used!

L342-352 tries to explain why the success in the first and last week is higher compared to the middle period. Authors refer to a difference between video and visual here too. See remark in previous paragraph on comparison between these two methods.

L336: “Thus it remains unclear why piglets did not use the urination area in these cycles”: the question rather is: why did some pigs (2) behave differently from the vast majority? I still don’t understand why some (2) pigs (in one batch) realised 40% correct urinations where others didn’t? Camera observation on the interaction between animals when going to the toilet, getting reward etc. should be informative.

In conclusion: especially due to the behavioural data, the study is worth publishing. Authors are challenged to go a bit deeper in the ethological evaluation of their results to enhance relevance for readers. Authors should be more honest that their aim (separation) failed and they should refrain from making advertisements in the M&M section for their farming co-author.

Details:

  • Line 58: is the 95% from the same source as the referenced 19%?
  • At first reading there is some misunderstanding that the description of the farm. For example, reference to ‘the animals’ (Line 105, Line 115) is not referring to the experimental animals, but to sows. Should be made more clear.
  • Number of pigs (24) is to be mentioned in the abstract
  • Line 145: provision of feed in the defaecation area and water in the urination area is OK when training, but should not be the common practice (separation of function areas!)

Author Response

Hello Reviewer 3,

Kind regards,

Michelle Tillmanns

Round 2

Reviewer 1 Report

DEAR AUTHORS,
thank you for responding to my comments.
Basically, in my opinion, there are only a few details left standing in the way of publication:

line 54:
Thank you for inserting the information after my comment. However, you still need to substantiate the statement with the reduced passage of feces at low animal densities with a literature source!

line 143:
You have not yet answered me whether the candies are approved as feed for pigs (probably not). Unfortunately, it is often not enough that foodstuffs are approved for humans, so you are not actually then automatically allowed to feed them to pigs. I would not see this so narrowly in the context of a trial, but you must indicate that for later practical use such "candies" must also be approved. Otherwise farmers or trial organizers are liable to prosecution (unfortunately).

Table 1:
Why was a diet for pregnant sows used? The protein content is actually too low for piglets, the fiber content is high. Although this ensures unproblematic rearing, problems could arise in practice with a different feed: if the piglets are later fed a "normal" feed with a higher energy and protein content as well as a lower crude fiber content, the intake of the additional candies can become a problem for the piglets (too much sweetness/energy --> intestinal health can be negatively influenced). Please make this connection clear in your paper so that imitators do not make a direct mistake with the feed composition.

Author Response

Dear Reviewer 1,

Have a wonderful Christmas for you and your family.

Kind regards,

Michelle Tillmanns

Reviewer 2 Report

I’m pleased to see the authors have addressed all my comments in great detail. Their answers and edits in the manuscript are satisfying and erase most of my previous concerns for publication. I still think there are a few points which could improve the quality of the manuscript. This are mainly points that the authors addressed very well in response to the reviewer but I believe adding some of these explanations to the manuscript would also benefit the reader.      

Please find below specific comments and suggestions I believe would further improve the manuscript.

L 77 delete second period (.) after end of sentence

L 91 - 93 “For successful toilet training, the animals must learn to recognise the excretory stimulus, suppress it until they reach the toilet, eliminate and then leave the toilet again” add reference (e.g. ref 20 (Dierksen et al., 2020))

after L99 It is important for the reader to know your hypothesis and what your claims are based on. You have done an excellent job in explaining this in your reviewer comments. I would suggest to add the following from your reply (or similiar) to the manuscript after L 99 –  “… we hypothesized that the placement of the drinker in this (the toilet area) should induce urination there immediately after drinking (as proposed in the literature; add reference). This first “quasi random” correct urinations after drinking should be rewarded (by providing a food reward) so that the animals establish a positive link between urination and this area, and intentionally will go there to urinate in future. To assess the success of this training, the number of urinations was observed in the urination area.”

L170 direct behavioural observations – following my questions why direct and video observations were undertaken, the authors have given a good justification in their response. I suggest that this is also made available to the reader by adding (as stated in the reviewer reply): “...direct observation was initially carried out in order to record events that could possibly be overlooked by the video surveillance, hence gain more accurate data”. Maybe add this after L181 and link to video analysis section state that another aim was to test whether information gained from the video and direct observation were comparable (i.e. justification for results section 3.2.)

L198 please add information that not all animals were observed via video analysis due to technical reasons. The authors have commented that is information was added in L 249 but I don’t find this in the edited manuscript (maybe the lines have moved?) L 198 (after “ i.e. a total of 16 pigs”) seems like an appropriate point to add this information.

L202 statistical analysis – thank you for clarifying your choice of statistical analysis. I would suggest to add at the end of the section (L213) that you run this exploratory analysis because there is currently no information how sex and weight impact urination behaviour, hence you conducted this analysis to add to the knowledge gap.

Regarding the discussion, I think the results are overall discussed in great detail. However, I advise to add few thoughts on limitations of their experimental design, i.e. regarding the lack of controls. Again, the authors have done a great job justifying this in response to the reviewer but I think the reader would also benefit to see this in the manuscript.

Author Response

Dear Reviewer 2,

Have a wonderful Christmas for you and your family.

Kind regards,

Michelle Tillmanns

Reviewer 3 Report

Above all: Compliments for the fast and predominantly adequate response to the earlier comments.

 Major: The manuscript still carries the sphere as if the authors are claiming to be inventors of some kind of pig toilet. They are not. Pigs are (as the authors also stated) sober almost from birth onwards. And readily able and willing to defaecate and urinate outside their lying- and activity area. ANIMALS is not the journal to expose patents or to claim innovations that are not theirs. Both are not professional. For this reason and for honest clarity, the patent text should not be “Dr. Kees Scheepens owns a patent with the patent number PCT/NL2014/050856 for the pig toilet used in the study” but: “The second author (KS) owns a patent with the patent number PCT/NL2014/050856 for the pig toilet equipped with the operant conditioning system for rewarding urination behaviour used in the study”.

Medior:

  • To be fully clear in the abstract, the conclusion should be more explicit regarding the aim of the (interesting!) innovation: Not: “Even if training was not successful,” But: “Even if training to spatially separate defeacation behaviour from urination behaviour,”
  • Comparison of direct observations and video recordings: I cannot follow the line of defence / explanation of the authors. Figure 3 does not provide such a comparison on an observation by observation-basis. Authors do understand that. I read in the M&M that there are periods / moments in which both observations were made. And this provides the opportunity to validate the video recordings in an objective way. If the authors do not want to do that, that is their choice.
  • On my plea to elaborate a bit more on the behavioural aspects, the authors reply “we do not have more data about the excretory behaviour which we could present here” . That is only partly true. From their data, they can depict whether pigs that go for defaecation often also go for urination. The degree of synchronisation of that behaviour is quite relevant for the thoughts behind their innovation. Although this is quite essential regarding their hypothesis, if the authors are unwilling to add ont figure/table, we’ll have to leave that choice to them. But to be honest, probably it would be the first time this was reported in a practical context for newly weaned pigs, and could deliver the most relevant citation-argument for the paper.

Minor:

Line 115: pigs were weaned at 6 weeks of age. Abstract: Piglets were weaned at 6-7 weeks

Line 126: Therefore, at 6 to 7 weeks of age, the experimental animals were weaned and brought over from the outdoor area to the indoor pen

  • Please be consistent in mentioning 6-7 weeks (in line 115)
  • Line 126 suggests that the experiment started directly after weaning, but is not explicit on this. Is that correct?

Conclusion

If the major remark is met and if the medior remarks are seriously considered, I would be happy to see this study published. The authors response addressed in the second and third bullet of the medior remarks suggest that they rather add the text than have to work on an extra table or figure. I regret that, but understand their point, and leave that to the editors. The minor remarks are rather advises that would aid in clarity.

Author Response

Dear Reviewer 3,

Have a wonderful Christmas for you and your family.

Kind regards,

Michelle Tillmanns
